# Comparative Study of Permanent Magnet, Conventional, and Advanced Induction Machines for Traction Applications

**Tayfun Gundogdu** [1,2]🔾, **Zi-Qiang Zhu** [2,*]🔾 **and Ching Chuen Chan** [3]

1   Department of Electrical and Electronic Engineering, Hakkari University, Zeynel Bey Campus, Keklikpinar, Pinarlar Cd., 30000 Hakkari, Turkey; tayfungundogdu@hakkari.edu.tr
2   Department of Electronic and Electrical Engineering, University of Sheffield, Mappin Street, Sheffield S1 3JD, UK
3   Department of Electronic and Electrical Engineering, The University of Hong Kong, Hong Kong, China; ccchan@eee.hku.hk
*   Correspondence: z.q.zhu@sheffield.ac.uk

**Abstract:** This paper investigates and compares the torque-generating capabilities and electromagnetic performance of advanced non-overlapping winding induction machines (AIM), conventional induction machines (CIM), and interior-permanent magnet (IPM) machines for electric vehicle (EV) applications. All investigated machines are designed based on the specifications of the Toyota Prius 2010 IPM machine. The steady-state and flux-weakening performance characteristics are calculated by employing the 2D finite element method and MatLab, and the obtained results are quantitatively compared. Furthermore, the torque-generating capabilities of three machines are investigated for different electric loadings, and the machine having the highest torque-generating capability is determined as AIM. Moreover, the major parameters affecting the torque-generating capability, such as magnetic saturation and magnet demagnetization, are examined in depth.

**Keywords:** electric vehicles; induction machine; interior permanent magnet machine; non-overlapping winding; saliency; torque capability

## 1. Introduction

Internal combustion engine vehicles are responsible for 21% of worldwide anthropogenic $CO_2$ emissions [1]. Making the transition of the global vehicle fleet to zero-emission vehicle technology is critical for decarbonizing road transportation and fulfilling the environmental and climate targets. Therefore, worldwide electric vehicle (EV) sales, including passenger cars, light trucks, and light commercial vehicles, reached 6.75 million units in 2021, corresponding to a 108% increase over 2020 [2]. It is very critical to choose the right electrical machine topology for EV applications in order to maximize efficiency, transient electromagnetic performance characteristics, flux-weakening capability, and cost. The worldwide five best-selling models in 2021 [3] and their electrical machine technologies are listed in Table 1. In addition, permanent magnet synchronous machines (PMSMs), particularly interior-permanent magnet (IPM) machines, are used in the world's top commercial EVs and hybrid electric vehicles (HEVs), including Toyota/Prius, Nissan/Leaf, BMW/i3, and numerous other vehicles. Other cars, on the other hand, including the BMW/X5, Renault/Kangoo, GM/EV1, Chrysler/Durango, and a few others, employ induction machines (IMs) [4–11]. Moreover, Tesla Motors Inc., one of the world's leading plug-in EV manufacturers, utilizes both IM (front) and IPM (rear) machines in its best-selling models, as seen in Table 1 [3]. In addition, Audi also utilizes the same traction topology in e-Tron models.

**Table 1.** Top selling EV models in 2021 and their machine technology [3].

| Top Selling Models | Machine Technology |
|---|---|
| Tesla/Model 3 | IM + IPM |
| Wuling/Hongguang Mini EV | PMSM |
| Tesla/Model Y | IM + IPM |
| Volkswagen/ID.4 | PMSM |
| BYD/Qin Plus PHEV | PMSM |

The common characteristics of electric machines designed for traction applications can be listed as [12,13]: (a) high starting torque; (b) high torque and high-power densities; (c) high efficiency across a broad speed range; (d) low torque ripple; (e) light weight; and (f) low cost.

Although there are many favorable characteristics of squirrel-cage IMs, such as robustness, relatively low-cost, good control dynamics, and mature manufacturing technology, having conductor bars in the rotor is its major disadvantage because the solid conductor bars of the IM cause additional joule losses in the rotor and hence result in a relatively lower efficiency compared to PMSMs. PMSMs, on the other hand, possesses superior advantages, such as high torque, high power, and relatively high efficiency. However, the high cost of NdFeB magnets has a significant impact on their popularity [8]. Moreover, there are a number of comparative studies on the performance comparisons of electrical machines employed in the traction applications in literature [4–28]. In these studies, the performance characteristics of IMs, PMSMs, switched and synchronous reluctance machines, PM assisted reluctance and brushless DC machines have been compared quantitatively. The squirrel-cage IMs have been noted as having established production technology and the ability to offer the required driving characteristics. PMSMs, on the other hand, have been noted as having better efficiency and torque density. Moreover, according to a comprehensive quantitative comparison study on electrical machines for traction applications, IPM machines have a higher power factor and efficiency than IMs, whereas IMs provide competitive performance characteristics at a lower cost and with better overloading capability. Wound field, reluctance, and variable flux synchronous machines have all been demonstrated to be less appealing for traction applications because of their poorer torque density and efficiency characteristics [20,22,27,28]. In addition, in order to improve the torque characteristics of PMSMs some unconventional methods have been proposed and compared with conventional electrical machines employed in traction applications [29,30].

Torque-generating capacities are investigated in this paper, in addition to previous research comparing the performance of IMs and PMSMs in the literature [5–9,12–14,19]. Therefore, the major goal of this study is to determine which machine is capable of producing higher torque under overload operating conditions and examine the underlying causes. In addition, this comparison incorporates an advanced induction machine (AIM) with non-overlapping windings, which was recently developed for EV/HEV applications [31].

This paper focuses on the analysis and quantitative comparison of electromagnetic performance and design characteristics of the IPM machine, CIM, and AIM operated under the same conditions and the same slot/pole number combinations. The major steady-state electromagnetic performance and flux-weakening characteristics are provided and discussed, with a particular focus on torque-generating capabilities. Moreover, since one of the AIM's most significant advantages is its comparatively small total axial length, the impact of stack length on steady-state performance is systematically investigated.

## 2. Research Method

### 2.1. Concept

This study presents FEA and electromagnetic performance comparisons of different types of electrical machines, namely PMSM and IM, employed in traction applications. Among the considered machines, the IPM and CIM are adopted by conventional integer

slot distributed windings (ISDWs), while AIM is adopted by advanced non-overlapping windings (ANWs).

### 2.2. Description of the Tool

An IPM machine, CIM, and AIM, having the same operating and design specifications as the Toyota Prius 2010 IPM machine, are modelled and analyzed by 2D, time stepping FEA. Accordingly, the flux-weakening performance characteristics are calculated by the Matlab tool by using the flux matrices obtained from FEA calculations.

### 2.3. Analysis Scheme

Various analyses, including transient, steady state, electric loading, and flux-weakening, have been conducted in this study. Transient and steady-state analyses have been conducted to reveal the rated performance characteristics such as electromagnetic torque, torque ripple, back-EMF and induced voltage waveforms, flux line and density distributions, and saturation factors. Electric loading analyses have been conducted to compare the torque-generating capabilities of the aforementioned machines. The flux-weakening characteristics are calculated to compare the torque and power versus speed curves and efficiency maps.

### 2.4. Research Results

Since the vehicle acceleration is of great importance, the torque-generating capabilities of different traction machines need to be revealed. In this study, apart from the comparison of the torque-generating capabilities, flux-weakening characteristics, torque ripple levels, and total active material costs have been compared quantitively. The results presented in this study show that the overall flux-weakening characteristics of IMs are comparable to those of IPM machines and because of the demagnetization of the magnets of IPM machine during overloading operating, the IMs deliver higher torque during acceleration or overloading modes.

## 3. Design Specifications

In order to evaluate the torque-generating capabilities, the performance characteristics of the studied machines are examined utilizing time-stepping 2D finite element method (FEM) for rated and various electric loading operations. In order to investigate the torque-generating capabilities, the electric loading will be varied from one to five times the rated current value. For a fair comparison and to achieve comparable results, the same operating conditions and geometrical parameters as shown in Table 2 are utilized.

For a fair comparison, the Toyota Prius 2010 IPM machine is directly adapted by applying the optimal specifications [21,32], and the IMs are designed with the same geometric dimensions and pole number. It is worth mentioning that Toyota Prius 2010 IPM machine has been used since all specifications and data for the Prius 2010 are fully available. However, other models are still not available because of the confidential issues of the company. On the other hand, some essential geometric and operating design specifications of electrical machines for traction applications can be found in [21–26]. Furthermore, the IPM machine and the CIM share the identical stator slot/pole number combination (48S/8P) and winding layout with 5-slot pitch single-layer windings. In addition, a double-layer non-overlapping 2-slot pitch winding topology is adopted for AIM having a 24S/8P combination.

IMs are optimized by using multi-objective global optimization via genetic algorithm, as presented in [33]. Two-dimensional cross-sectional views and the design specifications of the machines are shown in Figure 1 and given in Table 2, respectively. As shown in Table 2, all the machines have the same main operational and geometrical specifications. Moreover, the same iron core material whose core loss coefficients, namely hysteresis $k_h$, eddy current $k_c$, and excessive $k_e$ are given in the table, has been assigned as core material for stator and rotor parts of the machines. A strong grade of PM, whose remanence, relative permeability $\mu_r$, and coercivity $H_c$ are given in the table, is assigned for the IPM machine. Moreover,

because of the better heat and resistance characteristics of the copper over aluminium, copper has been assigned as material of the squirrel-cage [34].

**Table 2.** Specifications of Prius IPM machine and IMs.

| Parameters | IPM | CIM | AIM |
|---|---|---|---|
| S/(R or M) */P | 48S/16M/8P | 48S/52R/8P | 24S/26R/8P |
| Voltage limit (Vrms) | | 650 V $\times$ 85% $\times$ 2/$\pi$ | |
| Rated current (Apeak) | | 250 | |
| Number of coils per phase | 8 | 8 | 8 |
| Number of turns per coil | 11 | 8 | 11 |
| Number of series turn per phase | 88 | 64 | 88 |
| Fundamental winding factor | 0.966 | 0.966 | 0.866 |
| Number of parallel brunch | | 1 | |
| Slot fill factor | | 0.6 | |
| Phase resistance at 21 °C | 0.077 | 0.05612 | 0.0577 |
| Stack length (mm) | | 50.8 | |
| Stator parameters | | | |
| Outer diameter (mm) | | 264 | |
| Inner diameter (mm) | 161.9 | 185.85 | 184.8 |
| Tooth width (mm) | 7.55 | 8.45 | 11.52 |
| Slot opening (mm) | 1.88 | 1.88 | 5.8 |
| Slot height (mm) | 30.9 | 15.4 | 22/11 |
| Air-gap length (mm) | 0.73 | 0.4 | 0.4 |
| Rotor parameters | | | |
| Tooth width (mm) | — | 6.83 | 11.97 |
| Slot opening (mm) | — | 1.88 | 5.6 |
| Slot height (mm) | — | 14 | 20 |
| Magnet dimensions | 49.3 $\times$ 17.88 $\times$ 7.16 | — | — |
| Cage material | — | Copper | Copper |
| Iron grade | | DW310-35 | |
| $k_h$ | | 179.038 | |
| $k_c$ | | 0.375 | |
| $k_e$ | | 0.262 | |
| Magnet grade | NdFeB (N35) | — | — |
| $\mu_r$ | 1.05 | — | — |
| $B_r$ | 1.1 | | |
| $H_c$(kA/m) | −805.4 | — | — |

* R: Number of rotor slots. M: Number of permanent magnets. S: Number of stator slots.

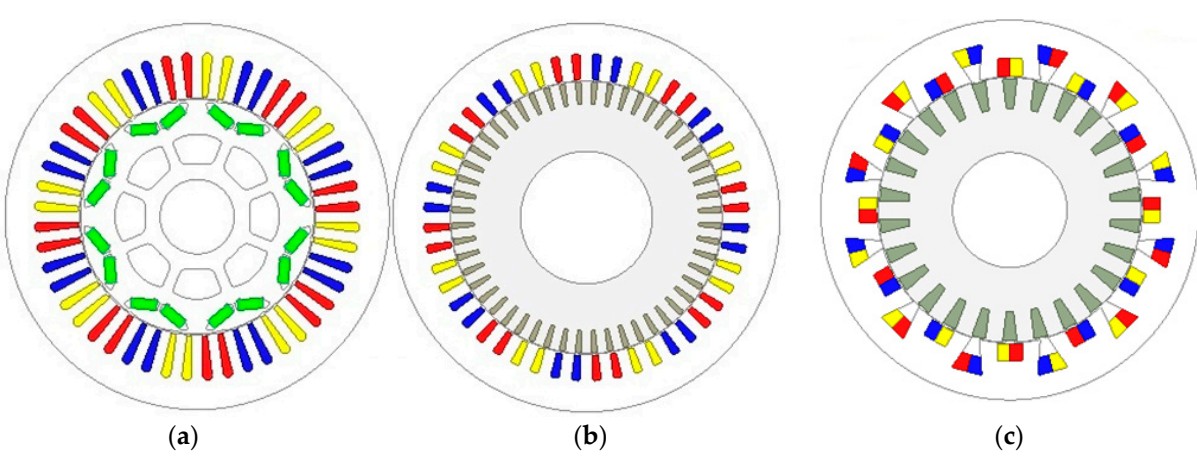

**Figure 1.** Two-dimensional views of the compared machines: (**a**) IPM (48S/16M/8P). (**b**) CIM (48S/52R/8P). (**c**) AIM (24S/26R/8P).

## 4. Performance and Active Material Cost Comparison

The performance characteristics of the IPM machine and IMs are determined using a *dq*-axis reference frame with the *d*-axis oriented with the rotor field [35]. The obtained results are presented in the following subsections.

### 4.1. Winding Factor and MMF Harmonics

The calculated winding factors and MMF amplitudes for 1-ampere-1-turn for 24S/8P (double-layer) and 48S/8P (single-layer) winding topologies are illustrated in Figure 2. As seen in Figure 2a, the fundamental winding factor and MMF amplitude of the 24S/8P topology are 10.35% lower than those of the 48S/8P topology. Therefore, it can be predicted that in order to obtain the same torque, the higher number of turns per phase is required for the 24S/8P topology under the operating conditions with the same excitation current. As seen in Figure 2b, the THD of the MMF of the 24S/8P topology is 49.68% higher than that of the 48S/8P topology. Hence, it can be expected that the rotor bar copper loss of 24S/8P topology will be considerably higher than that of the 48S/8P counterpart.

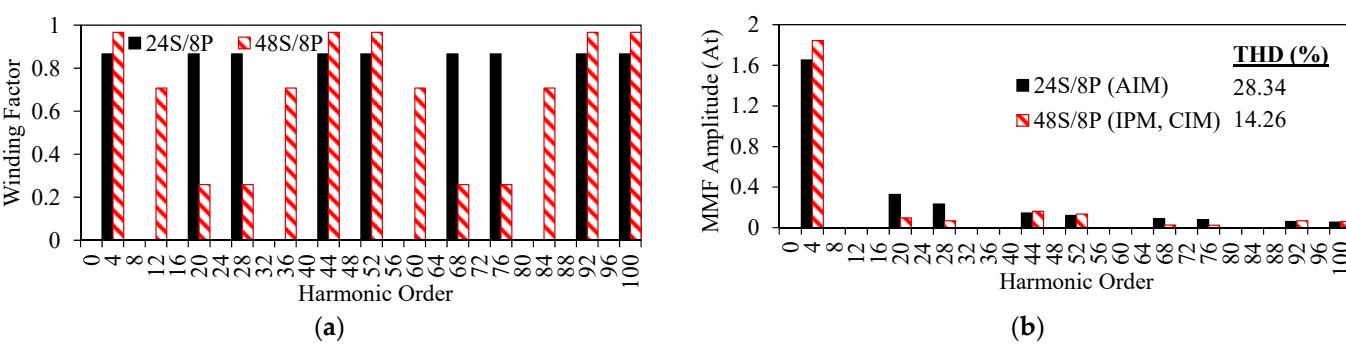

**Figure 2.** Comparison of winding factors and MMF harmonics of the considered machines. (**a**) Winding factors of harmonics. (**b**) MMF harmonics spectra for 1-turn and 1-ampere.

### 4.2. Induced Voltage and Back-EMF

The line back-EMF waveform for IPM machine and induced voltage waveforms for IMs, their harmonic spectra, and THD percentages are illustrated in Figure 3. As seen, the back-EMF waveform of the IPM machine is highly distorted. The possible reasons behind the distorted back-EMF waveforms can be the combined effect of stator slotting and the heavily saturated stator core. On the other hand, since the number of turns per phase and hence the ampere-turn magnitude at the same current is higher for the IPM machine, its back-EMF amplitude is higher than the IMs induced voltage values.

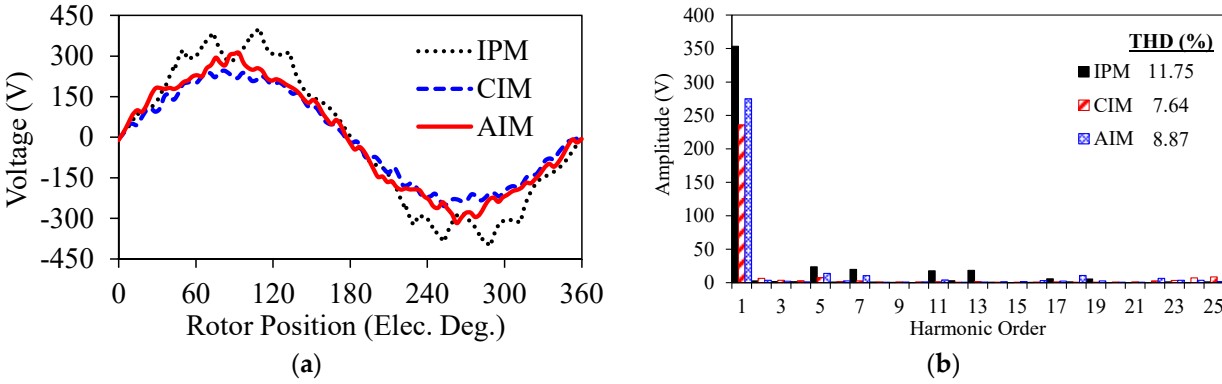

**Figure 3.** Comparison of back-EMF waveforms and their harmonic spectra. (**a**) Line Back-EMF and induced voltage waveforms. (**b**) Harmonic spectra of the back-EMF and induced voltage.

### 4.3. Flux Density and Flux Line Distributions

Figure 4 shows the flux density and flux line distributions of the considered machines. As expected, the flux density levels of the stator and rotor tooth parts are the highest. Among the designed machines, the CIM's averaged core saturation level is the lowest whilst the averaged core saturation of the IPM machine is the highest.

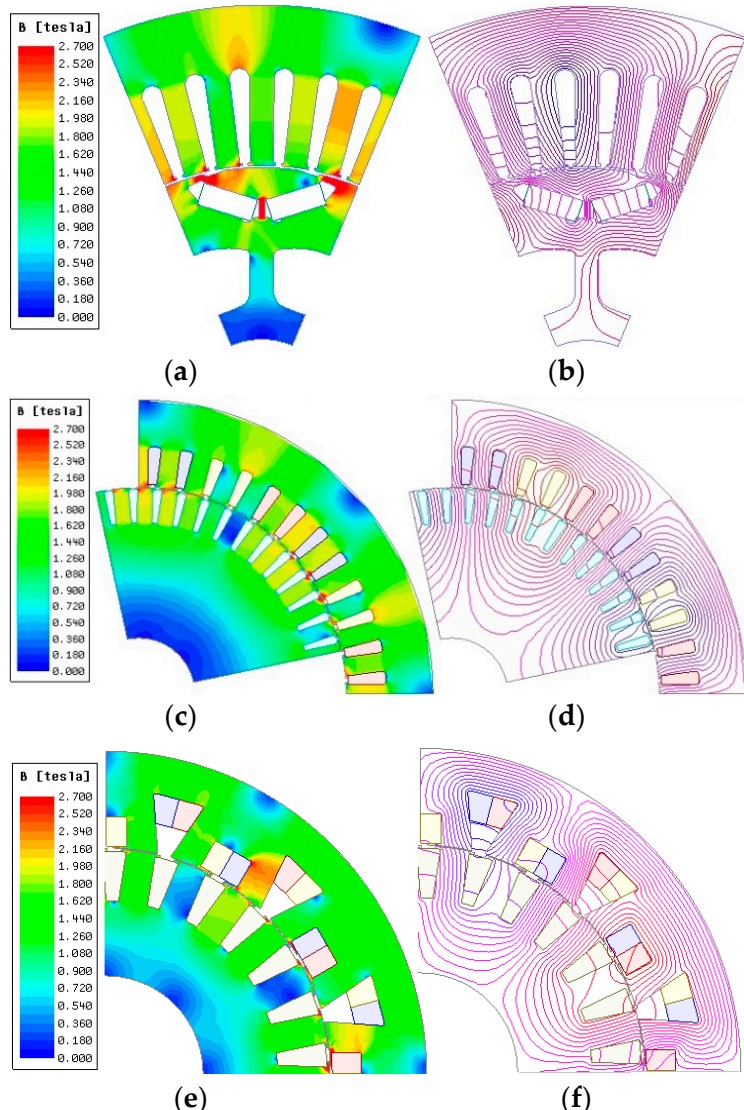

**Figure 4.** Flux density and flux line distributions of the machines. (**a**) IPM machine flux density. (**b**) IPM machine flux line. (**c**) CIM flux density. (**d**) CIM flux line. (**e**) AIM flux density. (**f**) AIM flux line.

### 4.4. Torque and Torque Ripple

The electromagnetic torque of the machines is calculated by using the expressions given between (1) and (3). As expressed in (1), there are two variable components contributing the torque of PM machines. The first variable term in (1) depends on the rotor excitation that is the PM flux $\psi_{PM}$ which depends on the properties of the PM material including size, maximum energy product $BH_{max}$, etc. and the second term, known as reluctance torque, depends on the saliency of the rotor which is determined by inductance components $L_q$ and $L_d$. $m$ is the phase number and $p$ is the pole pair number.

$$T_{em} = \frac{m}{2}p\left[\underbrace{\psi_{PM}I_q}_{Excitation} + \underbrace{(L_q - L_d)I_d I_q}_{Reluctance}\right] \tag{1}$$

The IPM machine's electromagnetic torque can be determined by modifying (1) with flux components as shown in (2). Furthermore, the electromagnetic torque of a squirrel-cage IM may be estimated using (3), which was developed for stator flux-oriented IM drives [36]. The superscript "*es*" in (3) indicates that the quantity is in the synchronous reference frame oriented to the stator flux.

$$T_{em\_IPM} = \frac{3}{2}p\left(\psi_d I_q - \psi_q I_d\right) \tag{2}$$

$$T_{em\_IM} = \frac{3}{2}p\left(\psi_d^{es} I_q^{es}\right) \tag{3}$$

For IPM machines and IMs, the current angle providing the maximum torque in motor operation mode has been determined to be 270° and 0° electrical degrees, respectively. Figure 5 illustrates the calculated electromagnetic torque waveforms and their harmonic spectra. As clearly seen in the figure, although all the machines have a similar average torque, the torque ripple percentage of the AIM is the highest. It is almost 2 times and 2.3 times higher than the CIM and IPM machines, respectively. Therefore, it is clear that special care should be taken during the design stage of the AIM. Consequently, an effective torque ripple reduction method involving utilizing u-shaped bridges on the rotor slots of the AIM is presented in [33]. As for IPM machines, numerous different ways exist to minimize the torque ripple, such as rotor, flux barrier, and magnet shape optimization, rotor skewing, harmonic current injection, stator slot optimization, and proper slot/pole number combination selection.

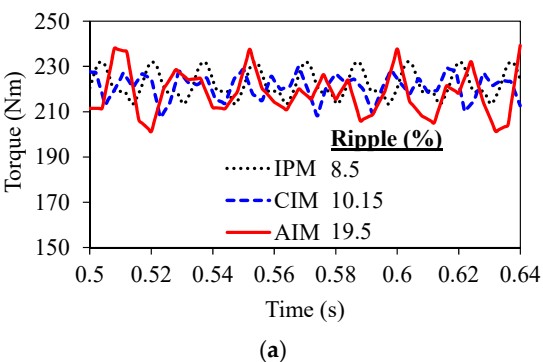 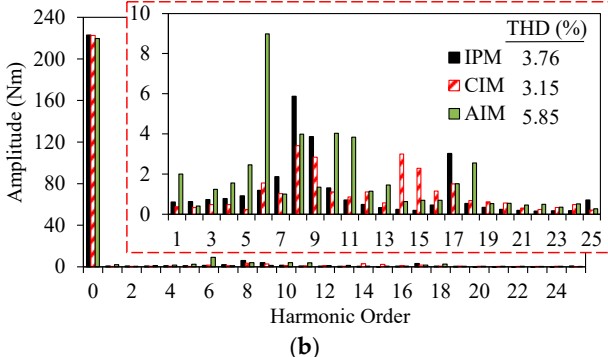

(**a**) 　　　　　　　　　　　　　　　　　　　　　(**b**)

**Figure 5.** Comparison of the torque waveforms and their spectra. (**a**) Electromagnetic torque waveforms. (**b**) Harmonic spectra of torque.

### 4.5. Flux-Weakening Characteristics

The machines' flux-weakening performances were computed using a combined numerical and analytical method described in [9,37]. The obtained torque-speed and power-speed curves of the machines are illustrated in Figure 6. Although IMs have remarkable flux-weakening performance in the constant torque region, their performance in the constant power region, particularly in the deep flux-weakening area, falls far short of that of IPM machines [9]. As seen in Figure 6a, all the machines have the same maximum output torque. Although IMs' characteristics are poorer than IPM machine at high-speed region, the maximum torque/power-speed characteristics of the IMs are not poorer than IPM machine. As clearly seen in Figure 6, in some speed regions, IMs show even better flux-weakening characteristics than the IPM machine. Therefore, in terms of flux-weakening characteristics, the IM can replace IPM with a slight sacrifice of torque in the very deep flux-weakening region. Moreover, it is also revealed that the AIM has poorer flux-weakening characteristics

than the CIM. The main underlying cause is that due to its lower fundamental winding factor, the AIM requires more turns to retain the average torque.

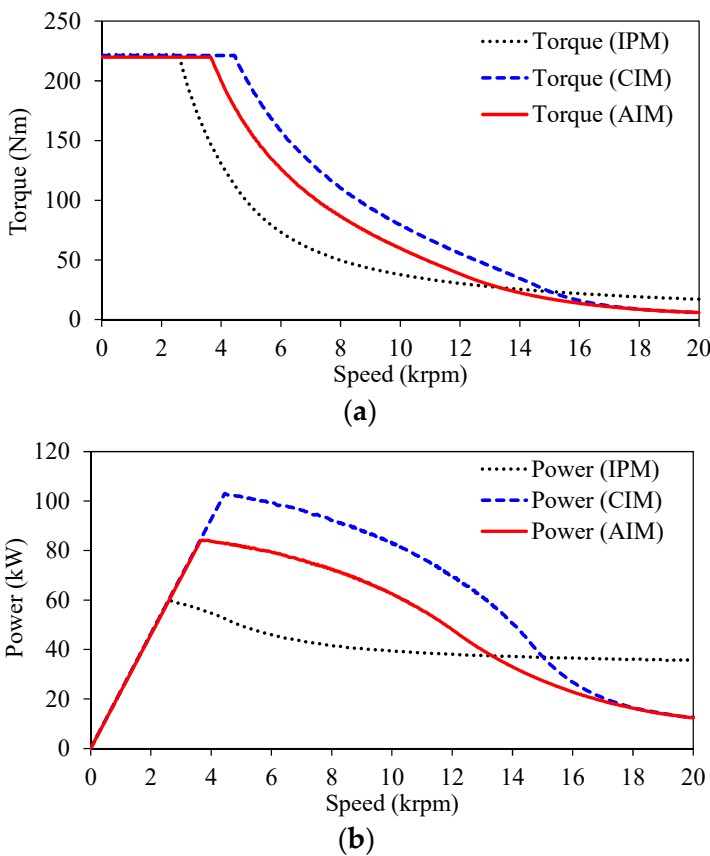

**Figure 6.** Flux-weakening characteristics. (**a**) Torque-speed characteristics. (**b**) Power-speed characteristics.

*4.6. Efficiency Maps*

Efficiency maps of the IPM machine and IMs are calculated for the 250Apeak current excitation operation considering the phase resistances calculated at 80 °C. In addition, mechanical loss $P_{mech}$ and additional losses $P_{add}$ including the friction, wind, and stray load loss have been taken into account as expressed in (4) and (5) [38], respectively.

$$P_{mech} = k_{m1}f_0 + k_{m2}f_0^2 \tag{4}$$

$$P_{add} = 0.01P_{out_1}\left(\frac{f_1}{f_n}\right)^{1.5}\left(\frac{I_n}{I_{NL}}\right)^2 \tag{5}$$

where $k_{m1}$ and $k_{m2}$ are the mechanical loss coefficients, $f_0$, $f_1$, and $f_n$ are the fundamental, working, and rated frequencies, respectively. $P_{out_1}$ is the working output power and $I_{NL}$ and $I_n$ are the no load and rated current amplitudes, respectively.

Calculated efficiency maps for the IPM machine, CIM, and AIM are illustrated in Figure 7. As clearly seen in Figure 7, the maximum efficiency is achieved at a different speed range for each machine. Moreover, the differences between the highest efficiencies are not significant. While the highest efficiency is achieved between 3–4 krpm for IPM, it is achieved between 4.5–7.5 krpm for CIM and 7–8 krpm for AIM. As seen from Figure 7c, the AIM has the highest efficiency at the deep flux-weakening region. The following are some of the most notable main findings:

- The IPM machine and CIM shows similar characteristics in terms of efficiency: lower efficiency at the lowest and highest speed regions;

- The efficiency of AIM at a lower speed is lower than those of the IPM machine and CIM. However, its efficiency at a higher speed is higher than those of the IPM machine and CIM.

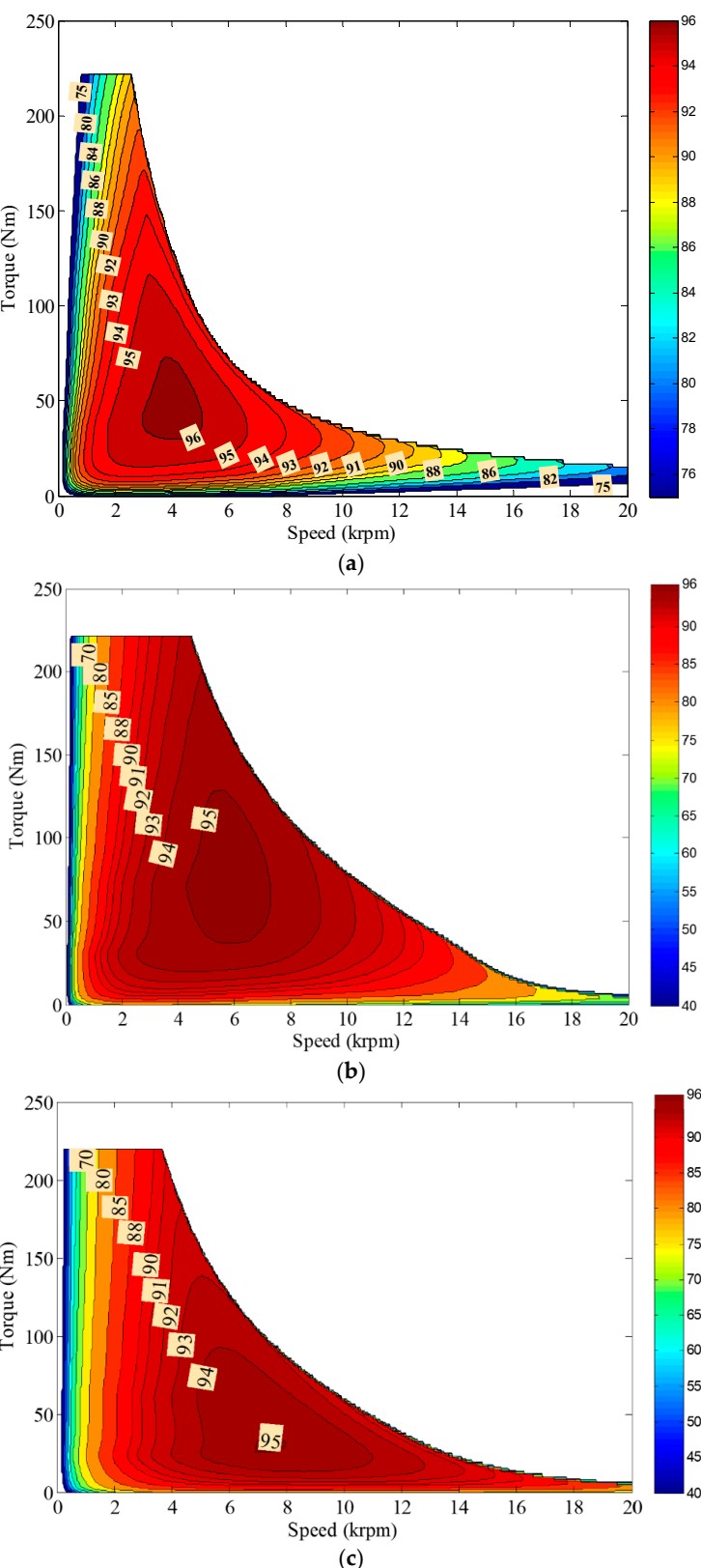

**Figure 7.** Comparison of the efficiency maps of the considered machines. (**a**) IPM machine. (**b**) CIM. (**c**) AIM.

### 4.7. Torque Production Capability

To be able to compare the torque-generating capabilities, the influence of electric loading on the torque is investigated. Figure 8 depicts the variation current angle of IPM and slip percentage of IMs that deliver the maximum torque. The current angle delivering the maximum torque for various peak current changes between 80° to 98°.

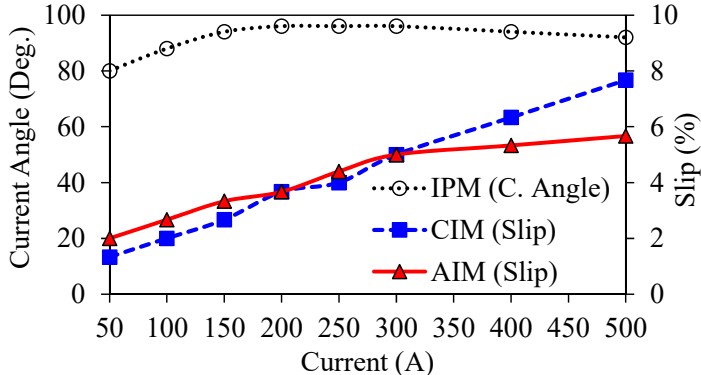

**Figure 8.** Variation of current angle (for IPM machine) and slip percentage (for IMs) with respect to peak current.

The current angle trend increases until peak inverter current limit (250Apeak) and then starts to decrease with the increasing current. However, for the IM, the slip, i.e., the difference between the stator and rotor magnetic fields, always increases with increasing peak current. Moreover, the AIM's overall slip, delivering the maximum torque, is lower than that of the CIM. The electromagnetic torque capability comparison of the machines is shown in Figure 9. Note that it is assumed that the conductors and isolation materials can operate safely under the excitation of twice the rated current operation, and the limitation of the current density is infinite. That means the thermal issues are ignored. As clearly seen in Figure 9, the increase in torque capability of IMs with peak current is much faster than that of an IPM machine. Furthermore, it is obvious from Figure 9 that the torque capability of the IPM machine is better for lower electric load operations than the rated current (250A). Once the electric loading starts to become higher than the rated current, the torque capability of the IMs becomes better.

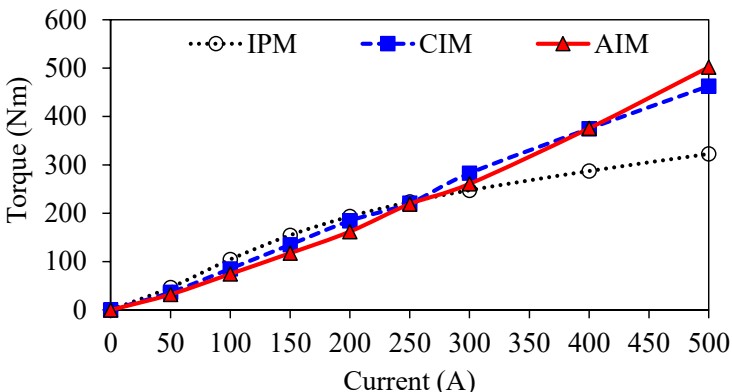

**Figure 9.** Comparison of torque production capabilities.

The torque ripple percentages for various electric loadings are compared in Figure 10. As seen, for the IPM machine, the higher the electric loading, the higher the torque ripple. On the other hand, for CIM, the higher the electric loading, the lower the torque ripple. For AIM, the torque ripple increases with the increasing electric loading until the rated current is reached. As previously explained, the electromagnetic torque of PM machines consists of PM and reluctance torque components. As seen in (1), the PM torque component is

proportional to the stator current, but the reluctance torque component is proportional to the square of the stator current. The reluctance torque component diminishes as the saliency ratio drops with increased electric loading. However, when the electric load increases, the impact of PMs becomes less prominent (PMs are getting demagnetized) as seen in Figure 11. Furthermore, the saturation caused by PMs may affect the increase in torque of the IPM machine. In fact, the IPM machine has PMs with constant flux magnitude. Even if the induced voltage is increased in the stator and the flux produced by the stator windings is increased with increasing excitation current, the flux produced by the PMs cannot be increased.

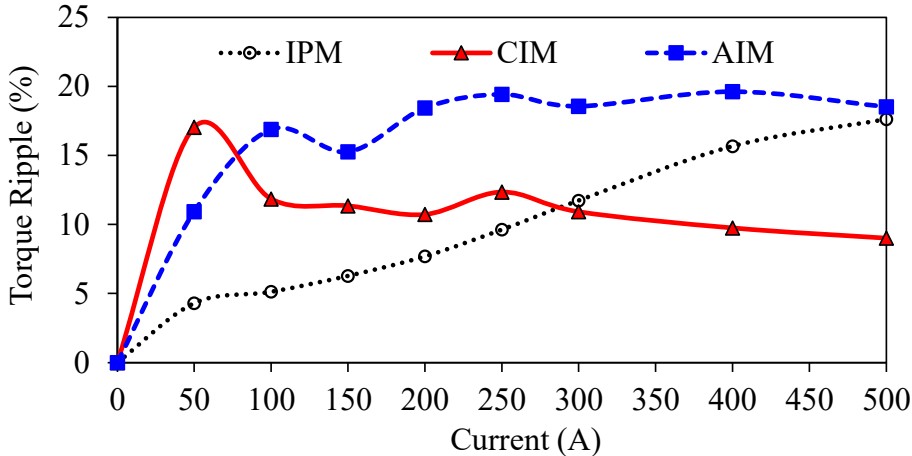

**Figure 10.** Comparison of torque ripple percentage for different electric loadings.

Essentially, as clearly seen in Figure 11, the flux generated by PMs dramatically reduces as the excitation current increases. This figure explains the underlying reason behind the lower torque generation of the IPM machine under overloading operating conditions. Once the IPM machine is loaded from 50% to 200%, the flux production capability of PMs reduces substantially. The flux produced by the stator windings dominates the flux produced by the PMs. In other words, PMs are temporarily demagnetized by the stator field. Therefore, since the flux density of the rotor core is much higher than the PM's flux density, quite a low flux can be produced by the PMs. Since the excitation and reluctance torque components of the IPM machine are decreased dramatically with increasing excitation current, it could not generate torque as high as IM (see Figure 9). For the IPM machine, the total flux is limited by PMs and saliency. However, for IM, with the increasing injected current, both the stator and rotor circuit's fluxes are increased since the rotor bar current will be increased by the increased excitation current. Therefore, since the total flux is increased, the obtained torque will also be increased. In theory, in comparison with PM machines, if the current density and heating issues are ignored, there is no torque limitation for the IMs. In order to estimate the saturation levels of the machines, the saturation factors for both of the machines have been calculated by using (6) and the variation of the calculated saturation factors with respect to electric loading is illustrated in Figure 12.

$$k_{sat} = 1 + \frac{MMF_S + MMF_R}{MMF_g} = 1 + \frac{H_S + H_R}{H_g} \tag{6}$$

where $MMF_S$, $MMF_R$, and $MMF_g$ are the magneto-motive force of stator, rotor, and air-gap, respectively, and $H_S$, $H_R$, and $H_g$ represent the surface integrations of flux intensity of the same regions. As seen in Figure 12, the saturation levels are similar, and they increase dramatically with increasing electric loading.

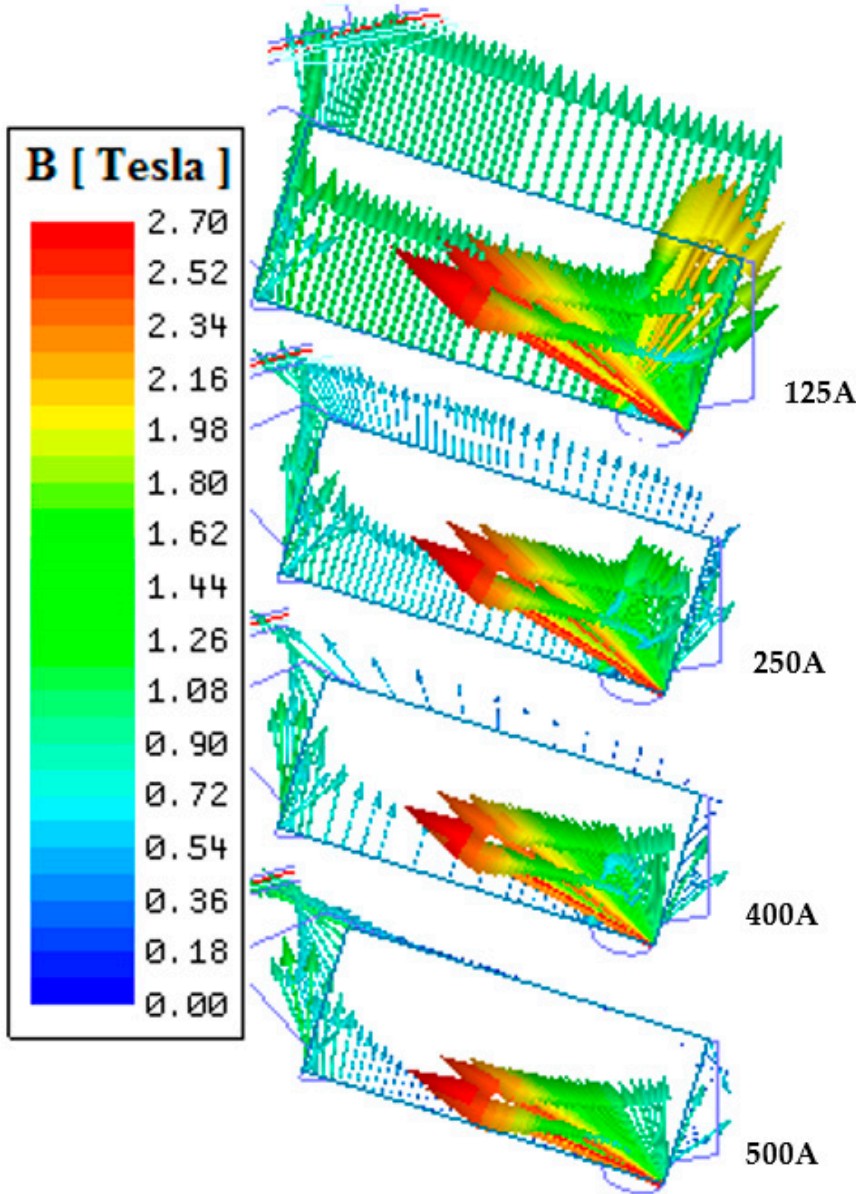

**Figure 11.** Flux density vectors of PMs for various electric loadings.

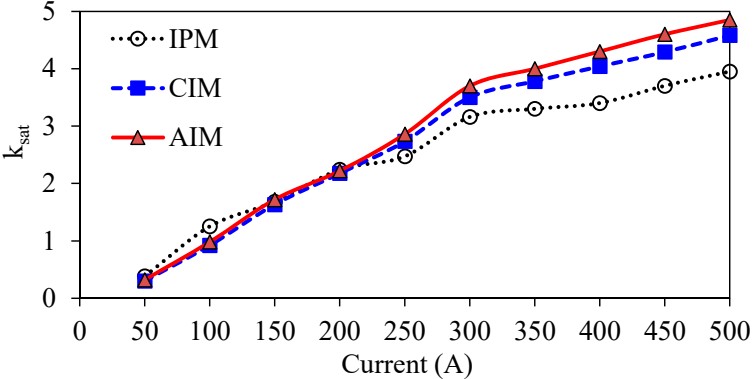

**Figure 12.** Variation of saturation factor with respect to electric loading.

### 4.8. Influence of Stack Lenght

In this section, in order to observe the performance of AIM designed with the same total axial length of the IPM machine and CIM, AIM1 and AIM2 have been designed, respectively. As presented previously, the AIM was designed with the same stack length as the Toyota Prius IPM machine. However, as shown in Table 3, once the AIM is designed with the same stack length as the Toyota Prius IPM machine, its total axial length is approximately 21.5% shorter than that of the IPM machine. In the same manner, it is 25% shorter than the CIM. Therefore, it is intended to investigate the performance of the proposed AIM, which is designed with increased stack lengths.

**Table 3.** Overall comparison of key results.

|  | IPM | CIM | AIM | AIM1 | AIM2 |
|---|---|---|---|---|---|
| $n_s$ | 88 | 64 | 84 | 60 | 60 |
| $\ell$ (mm) | 50.8 | 50.8 | 50.8 | 72.4 | 76 |
| $\ell_a$ (mm) | 97.5 | 101.1 | 76.66 | 97.5 | 101.1 |
| $R_{ph}$ ($\Omega$) | 0.077 | 0.0561 | 0.0577 | 0.0408 | 0.0371 |
| $T_{em}$ (Nm) | 222.38 | 220.75 | 219.15 | 218.38 | 220.45 |
| $\Delta T$ (%) | 8.45 | 10.15 | 19.63 | 19.5 | 19.8 |
| $s$ (%) | 0 | 3.33 | 4.4 | 3.2 | 3.33 |
| $P_{out}$ (kW) | 34.93 | 33.52 | 32.909 | 33.2 | 33.48 |
| $\eta$ (%) | 85.532 | 83.00 | 79.32 | 84.21 | 84.47 |
| $M_T$ (kg) | 22.7 | 25.22 | 24.279 | 31.02 | 32.7 |
| Cost (£) | 76.7 | 69.71 | 68.55 | 70.77 | 76.97 |
| $J_s$ (A/mm$^2$) | 21.77 | 28.52 | 28.76 | 23.92 | 21.6 |
| $J_r$ (A/mm$^2$) | — | 18.17 | 16.69 | 13.35 | 12.65 |

### 4.9. Comparison of Copper Losses

Figure 13 shows the calculated stator in-slot winding and end-winding, and rotor bar copper losses of the considered machines. As seen in Figure 13, almost the same stator slot copper losses are obtained except for the AIM. As clearly seen from the stator end-winding copper loss comparison, while the IPM machine, CIM, and AIM have almost the same value, the AIM1 and AIM2 have almost half the value due to the low number of turns per phase requirement. Note that, since the stack lengths of the AIM1 and AIM2 have been increased, their stator slot copper losses have also been increased. Therefore, it can be deduced that due to increased stack length, the number of turns and hence the stator copper loss is reduced. The rotor copper loss of the CIM is approximately half of the AIM's rotor copper losses because of the remarkably low (approximately half of the 24S/8P) winding MMF harmonics of 48S/8P with a 5 slot-pitch single-layer winding topology (see Figure 2).

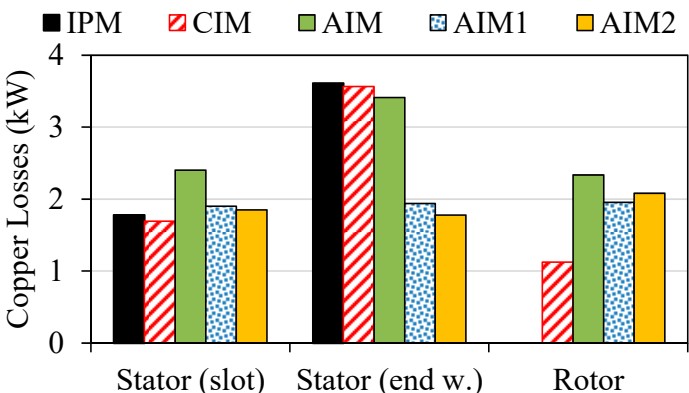

**Figure 13.** Comparison of copper losses including stator in-slot winding, stator end-windings, and rotor bars.

*4.10. Overall Comparison*

The electromagnetic performance and design characteristics of the IPM machine, CIM, AIM, and AIM1 and AIM2, which are designed with different stack lengths, are listed in Table 3. As can be observed, to be able to sustain the average torque, the number of turns of AIM1 and AIM2 are reduced while their stack lengths are increased. Therefore, it can be directly predicted that if the AIM topology is designed to have an equal total axial length with the IPM machine and CIM, the phase resistance would be reduced as shown in Table 3. As seen, if the AIM was designed with the same stack length as the IPM machine, its total axial length would be 21.5% and 25% shorter than the IPM machine and CIM, respectively, without sacrificing torque. On the other hand, if the AIM is designed to have the same total axial length as the IPM machine, the efficiency of the AIM can be increased by up to 6.5% without sacrificing the torque and output power. However, when the AIM is configured with same total axial length as that of the CIM, its efficiency is increased by 1.8% due to the increased output power.

The masses and the active material costs of each machine are calculated using the expression specified in (7) and (8) for IM and IPM machines, respectively,

$$M_{Tot_{IM}} = D_{W330}(A_{S_{core}} + A_{R_{core}})\ell + 2D_{cu_s}N_cA_{S_{coil}} \cdot \left[\ell + \tau_s - \left(\frac{b_{sw}}{2}\right)\right] + D_{cu_R}\left(RA_{Bar}\ell + 2A_{Ring}\ell_{Ring}\right) \tag{7}$$

$$M_{Tot_{PM}} = D_{W330}(A_{S_{core}} + A_{R_{core}})\ell + 2D_{cu_s}N_cA_{S_{coil}} \cdot \left[\ell + \tau_s - \left(\frac{b_{sw}}{2}\right)\right] + D_{PM}N_{PM}A_{PM}\ell \tag{8}$$

where $A_{Score}$, $A_{Rcore}$, $A_{Scoil}$, $A_{Bar}$, $A_{Ring}$, and $A_{PM}$ indicate surface areas of stator and rotor cores, stator coils, rotor bars, rotor end rings, and magnets respectively. $N_c$, $N_{PM}$, $R$, and $\ell_{Ring}$ are the total coil number, total magnet number, the number of rotor slots, and the axial length of ring, respectively. $D_{W330}$, $D_{Cu_S}$, $D_{Cu_R}$, and $D_{PM}$ indicate the mass densities of core, the stator copper, rotor bar copper, and magnet materials, respectively. In addition, $\ell$ is the stack length, $b_{sw}$ is the stator slot width, and $\tau_s$ is the average coil pitch. The cost of the raw materials is calculated using the most recent material price data from [39] and [40], as shown below:

- Copper: 7.354 £/kg—mass density: 7400 (kg/m$^3$);
- Steel: 0.44 £/kg—mass density: 8933 (kg/m$^3$);
- NdFeB35: 42.28 £/kg—mass density: 7520 (kg/m$^3$).

As shown in Table 3, raw material costs of all machines are similar. Therefore, considering the overall performance and flux-weakening characteristics, IMs can easily replace the expensive IPM machines. It is also revealed that if the stack length is increased, it is possible to reduce the stator and rotor current densities simultaneously.

**5. Conclusions**

In this study, the electromagnetic performance and design characteristics of the IPM machine, CIM, and AIM are quantitatively compared with particular reference to the torque-generation capability. The key findings of such a comparison have been summarized for the rated current operation condition and variable electric loading operations as follows:

- The overall flux-weakening characteristic of IMs are comparable to that of IPM machines;
- The flux-weakening characteristic of AIM are poorer than that of CIM;
- The overall efficiency of the IPM machine is higher than the CIM, and the difference between the maximum efficiency regions is 1.041% only;
- The efficiency of AIM is higher than CIM in deep flux-weakening regions;
- The torque ripple of the AIM is nearly 57% and 50% higher than that of the IPM machine and CIM, respectively, in the constant torque region;
- By extending the stack length without surpassing the total axial length of an IPM machine or CIM, it is feasible to significantly improve the output power and efficiency of AIM;

- It is also possible to reduce both the stator and rotor current densities simultaneously by extending the stack length;

Moreover, other important key findings related to the torque production capability and electric-loading operations can be summarized as follows.

The higher the electric loading level:

- The much faster the torque increases for IMs machines, the higher the torque levels become for the electric loading levels higher than the rated current (250Apeak), whilst it is vice-versa for the electric loading levels lower than the 250Apeak;
- The lower the torque ripple for the CIM, whilst it is higher for the IPM machine and the AIM;
- The higher the slip percentage for IMs;
- The higher the risk of demagnetization for the IPM machine.

It has been shown that non-overlapping winding topology is a very effective method, yielding several advantages, such as shorter axial length without compromising efficiency and torque, and simplicity in manufacturing. It has also been shown that the main disadvantages of the AIM topology are higher torque ripple and higher rotor copper loss due to very high amplitudes of MMF harmonics.

It has also been demonstrated that because of the AIM topology's vast design choices, smaller and relatively efficient IMs may be designed. If improved electromagnetic performance and efficiency are more significant design goals, it is possible to fulfill these criteria by extending the AIM's stack length without exceeding the entire axial length of the related CIM design.

**Author Contributions:** Conceptualization and methodology, T.G. and Z.-Q.Z.; software, T.G.; formal analysis, T.G., Z.-Q.Z. and C.C.C.; investigation, T.G. and Z.-Q.Z.; resources, Z.-Q.Z.; data curation, T.G.; writing—original draft preparation, T.G.; writing—review and editing, Z.-Q.Z. and C.C.C.; visualization, T.G.; supervision, Z.-Q.Z. and C.C.C.; project administration, Z.-Q.Z.; funding acquisition, Z.-Q.Z. All authors have read and agreed to the published version of the manuscript.

**Funding:** This research was partially funded by Valeo Powertrain Electric Systems, 94017 Créteil CEDEX, France.

**Institutional Review Board Statement:** Not applicable.

**Informed Consent Statement:** Not applicable.

**Data Availability Statement:** Not applicable.

**Conflicts of Interest:** The authors declare no conflict of interest.

## Abbreviations

| | |
|---|---|
| 2D | Two-Dimensional |
| AIM | Advanced Non-overlapping winding Induction Machine |
| ANW | Advanced Non-overlapping Winding |
| CIM | Conventional Induction Machine |
| $CO_2$ | Carbon Dioxide |
| DC | Direct Current |
| EMF | Electromotive Force |
| EV | Electric Vehicle |
| HEV | Hybrid Electric Vehicle |
| IM | Induction Machine |
| IPM | Interior-Permanent Magnet |
| ISDW | Integer Slot Distributed Winding |
| M | Magnet Number |
| MMF | Magnetomotive Force |
| NdFeB | Neodymium Iron Boron |

| P | Pole Number |
| --- | --- |
| PM | Permanent Magnet |
| PMSM | Permanent Magnet Synchronous Machine |
| R | Rotor Slot Number |
| S | Stator Slot Number |
| THD | Total Harmonic Distortion |

**Nomenclature**

| $A_{Scoil}$ | Surface Area of Stator Slot |
| --- | --- |
| $D_{CuR}$ | Mass Density of Rotor Bar Copper Material |
| $D_{CuS}$ | Mass Density of Stator Copper Material |
| $\ell_{Ring}$ | Ring Axial Length |
| $\ell_a$ | Total Axial Length |
| $A_{Bar}$ | Surface Area of Rotor Slot |
| $A_{PM}$ | Surface Area of Magnet |
| $A_{Rcore}$ | Surface Area of Rotor Core Lamination |
| $A_{Ring}$ | Surface Area of Rotor Ring |
| $A_{Score}$ | Surface Area of Stator Core Lamination |
| $D_{PM}$ | Mass Density of Magnet |
| $D_{W330}$ | Mass Density of Core Material (W330) |
| $H_R$ | Surface Integrations of Rotor Flux Intensity |
| $H_S$ | Surface Integrations of Stator Flux Intensity |
| $H_g$ | Surface Integrations of Airgap Flux Intensity |
| $I_{NL}$ | No Load Current Amplitude |
| $I_d$ | D-axis Current |
| $I_n$ | Rated Current Amplitude |
| $I_q$ | Q-axis Current |
| $I_q^{es}$ | D-Axis Current in Synchronous Reference Frame Oriented to Stator Flux |
| $J_r$ | Rotor Bar Current Density |
| $J_s$ | Stator Current Density |
| $L_d$ | D-axis Inductance |
| $L_q$ | Q-axis Inductance |
| $M_{Tot_{IM}}$ | Total Weight of IM |
| $M_{Tot_{PM}}$ | Total Weight of IPM Machine |
| $MMF_R$ | Rotor Magnetomotive Force |
| $MMF_S$ | Stator Magnetomotive Force |
| $MMF_g$ | Airgap Magnetomotive Force |
| $M_T$ | Total Weight |
| $N_{PM}$ | Number of Magnets |
| $N_c$ | Number of Coil |
| $P_{out_1}$ | Working Output Power |
| $P_{add}$ | Additional Power Loss |
| $P_{mech}$ | Mechanical Power Loss |
| $P_{out}$ | Rated Output Power |
| $R_{ph}$ | Phase Resistance |
| $T_{em}$ | Electromagnetic Torque |
| $b_{sw}$ | Stator Slot Width |
| $f_0$ | Fundamental Frequency |
| $f_1$ | Working Frequency |
| $f_n$ | Rated Frequency |
| $k_{m1,\,2}$ | Mechanical Loss Coefficients |
| $k_{sat}$ | Saturation Factor |
| $n_s$ | Serial Turn Number |
| $\tau_s$ | Average Coil Pitch |
| $\psi_{PM}$ | Magnet Flux |
| $\psi_d$ | D-axis Flux |

| $\psi_d^{es}$ | D-Axis Flux in Synchronous Reference Frame Oriented to The Stator Flux |
|---|---|
| $\psi_q$ | Q-axis Flux |
| $\Delta T$ | Torque Ripple |
| $\ell$ | Stack Length |
| $P$ | Pole Number |
| $R$ | Rotor Slot Number |
| $m$ | Number of Phases |
| $s$ | Slip |

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
