# Peer review of "Comparative Study of Permanent Magnet, Conventional, and Advanced Induction Machines for Traction Applications"

_wevj, doi:10.3390/wevj13080137_

Round 1

Reviewer 1 Report

Basic remarks of the reviewer:

The article presents the results of research on the possibility of generating torque and electromagnetic torquenet performance of advanced non-overlapping conventional induction winding (AIM) machines induction machines (CIM) and internal permanent magnet (IPM) machines for electric vehicles (EV). The study concerns the areas of engineering and electrical engineering that are important for researchers and fit into the subject of the journal and certainly deserve attention, in particular it concerns issues related to current problems related to electromobility and the development of drive units and the issue of generating torque so important in the process, e.g. vehicle acceleration.

However, the objectives of the study (especially since there are no hypotheses) are not well defined. The structure of the article is not fully transparent. The work is weakened by the lack of an innovative approach to the subject, i.e. drawing attention to the essence of the research problem and clearly emphasizing the authors' contribution to the development of research. The quality of the presented research is significantly limited due to the fact that it is reduced to one drive unit Toyota Prius 2010, de facto it is a vehicle in Hybrid technology, now this technology is outdated - hence the manufacturer has already introduced 4 generations of the drive and in 2023 the 5th generation enters the market. Thus, the title itself and the summary are not properly defined. There is a visible lack of the applied research method and its evaluation in real operating conditions. This approach does not allow for full inference - data analysis is based on behavioral models - there is no description of the limitations. The legibility of individual drawings is significantly limited 1.7. 11, and most of the drawings without detailed description are illegible. Moreover, there is no broader reference to similar research results of other scientists in the field of BEV, HEV, PHEV, MHEV, EREV, FCEV technologies concerning the same research problem. Each of the presented strategies-models should be described in detail along with their limitations, which, unfortunately, is missing, the mere indication that the Matlab tool was used is not sufficient in this case. Research should be conducted in such a way as to constitute a field for discussion with other researchers and be readable not only for researchers or the presenter - experts in this field, but also for others. The considerations presented in the conclusions do not fully result from the presented research. Literature itself is too scarce on this research topic. Each of the strategy-models should be described in detail, along with their limitations. Research should be conducted.

Detailed comments and areas requiring clarification:

1) The title of the article - should be clarified - as it is not precise in its current form - it is necessary to use the correct names for the type of drive and ask if the authors describe strategies, propose algorithm-based models, or are just trying to evaluate?

2) Abstract - should contain information on the context and background of the study, its purpose, procedures (selection of the drive unit, assumptions, measurements and methods, test dates, etc.), key results and main conclusions, research innovations and practical application.

3) Introduction - the introduction should clearly and concisely describe the background of the problem and the motives for undertaking the research - the whole should be supplemented with the description and characteristics of BEV, HEV, PHEV, MHEV, EREV, FCEV technologies.

4) The aim of the work and possible research hypotheses - it should be verified and redefined, emphasizing the innovation of research in the final part, including a description of the division of the article into individual sessions. Table 1 contains data on several drive units (type), but there is no description of the generation and year of production and the type of battery used. It is worth including a description of the newest engines produced in 2022.

5) Chapters 2,3- should be redrafted and constitute chapters under the title Research method together with the specification of individual chapters - Concept assumption, tool description, analysis scheme and the next chapter research results. At present, a serious scientific deficiency is the lack of a detailed list of individual subsections and their description, in particular the justification of the Matlab tool used and the choice of the drive unit.

6) The description of the type of drive unit that has been tested needs to be supplemented - technical parameters, year of production, method of operation, driving parameters, etc. whether the tests are only conceptual in Matlab or whether they were compared in the actual operating process. “- Description of the test procedure - the formulas and criteria used should be clear to a person who is not an expert in the field. It would be scientifically interesting to compare the parameters, for example, in different weather conditions. In particular, tests should be carried out. in such a way as to answer the questions posed in the article.

7) Formulas - require re-checking of their correctness and supplemented with an appropriate description,

8) The Results chapter - there is no description of the limitations in terms of the parameters tested, presented strategies or models. In the Summary - there was no reference to other researchers and research conducted, among others. by vehicle manufacturers and the evaluation of cost and economic parameters, e.g. with regard to analyzes of the selected drive unit - which certainly weakens the scientific analysis as a whole.

9) Conclusions and discussion - should refer to the presented considerations - reference to technological aspects and research conducted by other researchers dealing with this topic. Most of them are theses not supported by author's research. There are no indications of limitations resulting from further phased research and practical application.

10) No description of the abbreviations used, which significantly reduces the readability of the message.

11) References - References certainly need to be supplemented with the latest, related to the subject.

12) Used vocabulary - is not always scientific, but colloquial, which significantly weakens the scientific value of the article.

The reviewer would like to emphasize the authors' appreciation for taking up a research topic that is so rarely discussed and difficult, requiring the use of specialist equipment, including and time-consuming tests on a dynamometer or in real operating conditions. However, in its current form, the work requires the improvement of the scientific workshop and the research methodology used so that the presented considerations constitute the basis of a scientific article.

Reviewer 2 Report

The manuscript entitled Comparative Study of Permanent Magnet, Conventional, and 2 Advanced Induction Machines for EV/HEV Applications is well written and organised.  According to my revision it deserves to be published after completed the following revisions:

1.      In the introduction sections the authors should mention that also Audi utilizes  both types of motors IM and IPM

2.      Lines 205 and 206: the text should clearly refer to the equations 4 and 5

3.      The efficiency maps in Figure 7 should have the same maximum of efficiency 96 % for more clear comparison. In addition, the Figure caption should better describe the a, b and c figures

4.      Figure 11 is not clear and it should be enlarged and better described in the manuscript text

5.      Please explain what is the difference between the ksatn in the equation 6 and the ksat in the Figure 12?

Round 2

Reviewer 1 Report

Basic remarks

The recipient appreciates the authors' contributions at this stage. In the new version of the manuscript, the authors made some additions and responded to the reviewer's comments. However, there are still areas in the paper that, in the opinion, require little refinement or clarification.

At the outset, it should be noted that the role of the reviewer is to refer to the entire text sent, min. in terms of its evaluation according to specific scientific criteria. Therefore, the authors are aware that the comments or suggestions submitted by each reviewer may be of a different and different nature. On this, min. MDPI's review process is that several views on an article are taken into account. And the role of the reviewer is not to evaluate the scientific achievements of the authors or other reviews, but the scientific quality of the article and the presented research. The purpose of any comments and suggestions of the reviewer is to improve the scientific quality of the work and remove any shortcomings so that the scientific quality of the work is at the highest possible level. The Reviewer's care in this matter is dictated by the care for the prestige of the publishing house and special edition and, above all, by the authority of the authors. The reviewer is convinced that authors with such considerable achievements are aware of this.

Detailed suggestions.

1) Title of the article - electric vehicles differ in terms of propulsion solutions or methods of replenishing energy losses. The available literature distinguishes, among others, BEV (Battery Electric Vehicle), HEV (Hybrid Electric Vehicle), REEV (Range-Extended Electric Vehicle) and FCEV (Fuel Cell Electric Vehicle). That is why such a division is used among all researchers. In the reviewed article, the research focuses on the PHEV (plug-in hybrid electric vehicle) propulsion unit. Therefore, the title of the work, abstract and all considerations, should refer to the correct nomenclature. The reviewer is convinced that authors with such authority are aware of this and will consider a change.

2) Research method - division into sections. In scientific articles printed in the MDPI, in particular research or experimental, for many years a scientific scheme and a scientific title have been adopted, in which an important element of the work is the division of the article into individual sections and one of its elements is, among others Research method with the specification of individual sub-chapters - Concept assumption, description of the tool, analysis scheme and research results for the next chapter. The description of the procedure should be clear to a person who is not an expert in the field, such as the authors and the reviewer. Recenent is also convinced that authors with such scientific achievements are also aware of this and will consider changing the current system.

3) When it comes to the availability of data for individual drive units of vehicles from the BEV group, I encourage the authors to read the rich literature of European studies, for example in the pages of MDPI. They are available to researchers and individual research institutes, including VW or MERCEDES groups.

4) Regarding the data in Table 1, Receznent encourages you to read these data https://www.caranddriver.com/features/g36278968/best-selling-evs-of-2021/

https://www.statista.com/statistics/960121/sales-of-all-electric-vehicles-worldwide-by-model/ The reviewer is confident that the authors will review these results and will amend them if necessary. Data for the first half of 2022 are also available.

5) Chapter Performance and Cost Comparison - the phrase cost is an economic category and is expressed in a monetary measure, which is clearly stated in the professional literature on the subject.

The reviewer is also convinced that authors with such authority are aware of this. Therefore, it is necessary to change the title or refer to this category in the content of this chapter.

5) Including abbreviations and descriptions of formulas in the final part of the work. In the opinion of the reviewer, it will improve the clarity of the whole presented considerations.

6) References- supplementing the literature with considerations of European researchers on this research topic will certainly, based on the opinion of both readers and other researchers, will contribute to confirmation of the thorough analysis of the literature on the subject by the authors of the article.

Once again, the Reviewer would like to point out that the suggestions made are intended to increase the scientific value of the presented considerations and for the benefit of the authors of the article. The reviewer is convinced that the authors are aware of the purposefulness of such a procedure.
